The role of fragmentation and landscape changes in the ecological release of common nest predators in the Neotropics

Cove Michael V. 1 3 mvcove@ncsu.edu
Spínola R. Manuel 2
Jackson Victoria L. 1
Saénz Joel C. 2
1 Department of Biology and Agriculture, University of Central Missouri , Warrensburg, MO , USA
2 Instituto Internacional en Conservación y Manejo de Vida Silvestre, Universidad Nacional , Heredia , Costa Rica
Xu Jianhua
3 Current affiliation: Department of Applied Ecology, North Carolina State University, Raleigh, NC, USA

Electronic publication date: 2014 Jul 1
Publication date: 2014
Volume: 2
Electronic Location ID: e464
Received 2014 Mar 31; Accepted 2014 Jun 12
Copyright: © 2014 Cove et al.
Copyright year: 2014
Copyright holder: Cove et al.
License: This is an open access article distributed under the terms of the Creative Commons Attribution License, which permits unrestricted use, distribution, reproduction and adaptation in any medium and for any purpose provided that it is properly attributed. For attribution, the original author(s), title, publication source (PeerJ) and either DOI or URL of the article must be cited.
License URL: https://creativecommons.org/licenses/by/4.0/

Keywords: Camera traps, Carnivores, Coati, Fragmentation, Mesopredator release, Neotropics, Nest predators, Pineapple production, Tayra, Raccoon

Funding: Funding for this research was provided by the Universidad Nacional de Costa Rica and the University of Central Missouri International Center. The funders had no role in study design, data collection and analysis, decision to publish, or preparation of the manuscript.

==============================
Loss of large mammalian carnivores may allow smaller mesopredators to become abundant and threaten other community members. There is considerable debate about mesopredator release and the role that other potential factors such as landscape variables and human alterations to land cover lead to increased mesopredator abundance. We used camera traps to detect four mesopredators (tayra, Eira barbara; white-nosed coati, Nasua narica; northern raccoon, Procyon lotor; and common opossum, Didelphis opossum) in a biological corridor in Costa Rica to estimate habitat covariates that influenced the species’ detection and occurrence. We selected these mesopredators because as semi-arboreal species they might be common nest predators, posing a serious threat to resident and migratory songbirds. Pineapple production had a pronounced positive effect on the detectability of tayras, while forest cover had a negative effect on the detection of coatis. This suggests that abundance might be elevated due to the availability of agricultural food resources and foraging activities are concentrated in forest fragments and pineapple edge habitats. Raccoon and opossum models exhibited little influence on detection from habitat covariates. Occurrence models did not suggest any significant factors influencing site use by nest predators, revealing that all four species are habitat generalists adapted to co-existing in human altered landscapes. Furthermore, fragmentation and land cover changes may predispose nesting birds, herpetofauna, and small mammals to heightened predation risk by mesopredators in the Neotropics.

Introduction

Large carnivores receive substantial attention from the scientific community due to their charismatic status, their importance as keystone species in maintaining community structure, and their susceptibility to extirpation from habitat loss (Morrison et al., 2007). Due to the loss of large carnivores from many systems, medium-sized carnivores and carnivorous marsupials (collectively known as mesopredators) have recently gained more attention (Roemer, Gompper & Van Valkengurgh, 2009) because of their potential threat to migratory song birds (Crooks & Soulé, 1999; Donovan et al., 1997; Elmhagen & Rushton, 2007; Garrott, White & Vanderbilt White, 1993). The most commonly cited hypothesis for the increased abundance of mesopredators is the Mesopredator Release Hypothesis (MRH; Crooks & Soulé, 1999), but there is much debate that remains over the process of mesopredator release (Cove et al., 2012a; Gehrt & Clark, 2003; Elmhagen & Rushton, 2007). The MRH has support within some families, most notably the Canidae, where larger canids drive the population dynamics and habitat associations of smaller canids (Donadio & Buskirk, 2006; Gehrt & Clark, 2003). However, interspecific killing and interference competition are less common among different families (Donadio & Buskirk, 2006).

The top predators in Central America are jaguars (Panthera onca), pumas (Puma concolor), and, coyotes (Canis latrans; Cove et al., 2012b; Wainwright, 2007). All three predators partake in interspecific killing and may affect the distribution and habitat associations of smaller mesopredators (Donadio & Buskirk, 2006; Palomares & Caro, 1999). The large cats are often hunted due to cattle depredation and are rare. Coyotes are recent invaders and, because they prefer disturbed open habitat, are seemingly rare in many intact forested areas. Mesopredators are common in Central America which may result from the rarity of the top predators, thus, supporting the MRH and a “top down” view of their release. Another plausible explanation is that mesopredators are better adapted to coexisting with humans in disturbed habitats and, as omnivores, are able to supplement their diets with agricultural resources via a “bottom up” release (Elmhagen & Rushton, 2007; Garrott, White & Vanderbilt White, 1993; Roemer, Gompper & Van Valkengurgh, 2009).

Several studies in the United States attempted to model relative abundance of mesopredators as functions of landscape and local habitat variables and predict predation risk for forest-nesting birds (Crooks & Soulé, 1999; Dijack & Thompson, 2000; Donovan et al., 1997). These studies determined that mesopredator abundance, activities, and hence nest predation increased in fragmented areas and within forest edge habitats, particularly surrounding agricultural lands. However, no similar studies have examined mesopredator occurrence in Central America.

This study integrated data collected from camera traps and the occupancy modeling framework developed by MacKenzie et al. (2005) and MacKenzie et al. (2006) in order to examine habitat associations of four common mesopredators from three different families in a fragmented biological corridor in Costa Rica. We selected tayra (Eira barbara), white-nosed coati (Nasua narica), northern raccoon (Procyon lotor), and common opossum (Didelphis opossum) as the species of interest because they are common, adapted to human presence, and are important nest predators due to their semi-arboreal nature.

Methods

Study site

The San Juan–La Selva Biological Corridor is the northernmost portion of the Mesoamerican Biological Corridor in Costa Rica linking the Indio-Maíz Biological Reserve of southeastern Nicaragua to the Braulio Carrillo National Park of central Costa Rica. At its northern extent, the corridor also contains the proposed Maquenque National Park, which is the least fragmented area in the region. Although deforestation of primary forest still occurs within the corridor, government incentives (Forestry Law no. 7575) have encouraged reforestation and tree plantations (Morse et al., 2009). Most of the land within the corridor is privately owned with some reserves and lodges established to mitigate small scale agriculture, yet large scale pineapple plantations and cattle operations continue to expand in the corridor and surrounding landscape, particularly in the southern region (Fagen et al., 2013). We selected 16 survey sites to be representative of the land cover in and around the corridor; selection was loosely based on accessibility, forest patch size, and coverage along the entire corridor (Fig. 1). All forest sites were located on eco-lodge forest reserves, tree plantations, cattle ranches and agricultural plantations. In an effort to ensure independence among forest sites, we selected sites a minimum of 2 km apart.

Figure 1 Map of camera trap survey locations and the forest cover (including primary, secondary, and tree plantations) within the San Juan – La Selva Biological Corridor and its relative location in Costa Rica.

Sites located outside of the corridor were located in isolated forest fragments except the furthest south, which was located in Braulio Carillo National Park. The inset diagram shows the relationship that we examined of the effects of land cover change on nest predators and the apparent relationships to large predators and Neotropical birds.

Camera trapping

We surveyed fourteen forest sites over two field seasons (July–August 2009 and June–August 2010), while two additional sites were surveyed from October–November 2009. Information was lost from one of these sites and was excluded from further analysis. To avoid the pitfalls of using a single camera as a defined “site” representative of an entire forest (Efford & Dawson, 2012), we decided to aggregate several traps in arrays at each site. Arrays consisted of a central camera station and three additional camera stations surrounding the central station spaced at >250 m apart, for a total of four cameras in the 2009 surveys. Cameras were arranged in an array of six spaced >250 m apart in the 2010 surveys. Previous research suggested that although this resulted in variable trapnights among sites, there were no strong differences in detection as result of the varying numbers of cameras (Cove et al., 2013). Each camera station consisted of a remotely triggered infrared camera (Scout Guard SG550; HCO Outdoor Products, Norcross, GA, USA) or a remotely triggered flash camera (Stealth Cam Sniper Pro Camera 57983; Stealth Cam, LLC, Grand Prairie, TX, USA) secured to a sturdy tree 0.25–0.5 m off the ground. The camera was directed at an opposing tree, 3–4 m away, baited with a secured can of sardines 1–1.5 m off the ground. Although other camera trap studies set cameras along human trails and roads (Tobler et al., 2008), we avoided areas of high human use due to threat of theft focusing on animal game trails. Trail cameras were left at each site for 24–38 days and checked weekly for rebaiting and battery changes.

All of our research was in accordance with the guidelines established by The American Society of Mammalogists (Gannon & Sikes, 2007). The camera trapping protocol was approved by the University of Central Missouri Institutional Animal Care and Use Committee (IACUC–Permit No. 10-3202).

Habitat variables and analysis

Using ArcGIS 10.0 (ESRI, Redlands, CA, USA), we overlaid camera trap locations onto a digitized land use-land cover map. We created a 1-km radius buffer at each site using a central point among the cameras to measure landscape covariates. Habitat covariate selection was based on known ecology of the mesopredators and factors that might affect their detection and occurrence (Gompper, 1995; Lotze & Anderson, 1979; McManus, 1974; Presley, 2000; Wainwright, 2007).

We measured the distance to the nearest village, creating an index of human presence and/or disturbance. Forest cover is the percentage of buffer covered by primary and secondary forest and tree plantations. Because pineapple plantations are prevalent in the region, we used a binomial covariate to indicate this land use within each site buffer. The proposed Maquenque National Park is also within the northern extent of the corridor and we used a similar binomial covariate to denote sites as within or outside the proposed park boundaries. The final covariate was the total number of patches within each site buffer, which is an index of habitat heterogeneity and habitat fragmentation. We standardized all continuous covariates to z scores for analysis, but performed no other transformations (Long et al., 2011).

We combined all mesopredator photos from both field seasons to organize and manage binary detection histories (1 = detected, 0 = not detected). We partitioned detection histories into five day blocks for a maximum total of seven repeat surveys per species per site. We used the detection histories and habitat covariates within a single-season occupancy model in program PRESENCE 2.4 (Hines, 2009). Although the data were collected over two field seasons, we did not resample any sites. This analysis refers to Ψ as “site use” as opposed to “occurrence”, so grouping of the field seasons does not violate any of the assumptions of the modeling process (MacKenzie et al., 2005).

Given our data, we developed six relatively simple a priori models for each species (Table 1), including a global model, to estimate the influence of habitat covariates on detection probabilities in the individual mesopredator detection models. Although detection probability is often considered a nuisance parameter, there is an apparent relationship between detection probability and local abundance because as local abundance increases the probability of detecting a species will increase making it a parameter of interest in our study (O’Connell & Bailey, 2011). We did not use a seasonal covariate because all surveys were conducted during the rainy season. We then used the covariates that contained high model support and had strong effects on detection as a constant covariate set in the subsequent occurrence models (Long et al., 2011). For the occurrence models, we used seven a priori models (Table 2).

Table 1 A priori hypotheses regarding detection.

Descriptions and expected direction of a priori detection (p) models for mesopredators from camera trap surveys in the San Juan - La Selva Biological Corridor, Costa Rica, 2009–2010.

Hypothesis	Model	Model structure	Expected result	
No habitat covariates affect detection	p(.)	β 0	–	
Mesopredator abundance and foraging increase in close proximity to
villages so as distance to village increases detection decreases	p(dist)	β0 + β1(dist)	β1 < 0	
Habitat heterogeneity and fragmentation increase forest edge and
lead to higher foraging and detection	p(tnp)	β0 + β1(tnp)	β1 > 0	
Pineapple production provides food resources and increases
abundance and detection	p(pina)	β0 + β1(pina)	β1 > 0	
Increasing forest cover will have a negative effect on detection
because activities will be less concentrated	p(for)	β0 + β1(for)	β1 < 0	
Distance to village, habitat heterogeneity, pineapple production, and
forest cover all affect detection	p(global)	β0 + β1(dist) + β2(tnp)
+  β3(pina) + β4(for)	β1 < 0, β2 > 0,
β3 > 0, β4 < 0	

Table 2 A priori hypotheses regarding occurrence.

Descriptions and expected direction of a priori occurrence (Ψ) models for mesopredators from camera trap surveys in the San Juan - La Selva Biological Corridor, Costa Rica, 2009–2010.

Hypothesis	Model	Model structure	Expected result	
No habitat effects on occurrence	Ψ(.)	β 0	–	
Negative effect on occurrence within Maquenque National Park	Ψ(Maq)	β0 + β1(Maq)	β1 < 0	
Negative effect on occurrence as distance to village increases	Ψ(dist)	β0 + β1(dist)	β1 < 0	
Positive effect on occurrence as habitat heterogeneity increases	Ψ(tnp)	β0 + β1(tnp)	β1 > 0	
Negative effect on occurrence as forest cover increases and provides
habitat for larger predators	Ψ(for)	β0 + β1(for)	β1 < 0	
Positive effect on occurrence with pineapple production present	Ψ(pina)	β0 + β1(pina)	β1 > 0	
Maquenque National Park, distance to village, habitat heterogeneity,
forest cover, and pineapple production all affect occurrence	Ψ(global)	β0 + β1(Maq) + β2(dist) + β3(tnp) + β4(for) + β5(pina)	β1 < 0, β2 < 0,
β3 > 0, β4 > 0, β5 > 0	

We determined the best approximating models based on the Akaike Information Criterion corrected for small sample size (AICc) and Akaike weights (ωi). To evaluate model fit, we performed 10,000 simulated parametric bootstraps for the global model (all covariates) to determine if there was evidence of overdispersion (Burnham & Anderson, 2002). We considered all models contained within the 90% CI (∑ωi > 0.90) to have substantial support as the top-ranking models (Burnham & Anderson, 2002).

Results

From 2,286 camera trapnights, we obtained 23 independent photographs of tayras (10.06 photos/1000 trapnights), 33 photos of coati (14.44/1000 trapnights), 7 photos of raccoons (3.06/1000 trapnights), and 23 photos of opossums (10.06/1000 trapnights). At least one nest predator species was detected at every site, but only one site had detections of all four species. There was no evidence of overdispersion and we evaluated all models by their AICc and their Akaike weights.

Detection covariates affected each species differently (Table 3). Pineapple production had high model support (∑ωi = 0.75) and a strong positive influence on detection probability for tayras and was used as the constant detection covariate in subsequent occurrence models. Forest cover had high model support (∑ωi = 0.67) and a negative influence on detection probabilities for coatis and was used as the constant detection covariate in coati occurrence models. Raccoon and opossum models contained minimal support for habitat covariates influencing detection and we used a constant detection probability in the eventual occurrence models.

Table 3 Results for detection.

Selected top models and untransformed coefficients of habitat variable effects on detection probability (p) for mesopredators from camera trap surveys in the San Juan - La Selva Biological Corridor, Costa Rica, 2009–2010.

				Untransformed coefficients of covariates (SE)	
Species model	Δi	ωi	K	Intercept	Pineapple	Total number of patches	Distance	Forest	
Tayra									
p(pina)	0.00	0.611	3	−2.695 (0.662)	2.211 (0.743)	–	–	–	
p(global)	2.97	0.138	6	−3.646 (0.837)	2.626 (0.918)	1.016 (0.423)	0.659 (0.338)	−0.303 (0.502)	
p(tnp)	3.38	0.113	3	−1.311 (0.465)	–	0.701 (0.368)	–	–	
p(.)	3.89	0.087	2	−0.807 (0.355)	–	–	–	–	
Coati									
p(for)	0.00	0.667	3	−0.214 (0.291)	–	–	–	−0.784 (0.330)	
p(.)	2.93	0.154	2	−0.329 (0.279)	–	–	–	–	
p(pina)	4.26	0.079	3	−0.705 (0.409)	0.763 (0.567)	–	–	–	
p(dist)	4.65	0.065	3	−0.313 (0.279)	–	–	−0.322 (0.271)	–	
Raccoon									
p(.)	0.00	0.383	2	−1.157 (0.599)	–	–	–	–	
p(tnp)	1.15	0.216	3	−1.793 (0.805)	–	0.691 (0.471)	–	–	
p(dist)	2.15	0.131	3	−1.035 (0.570)	–	–	0.819 (0.747)	–	
p(pina)	2.46	0.112	3	−0.444 (0.915)	−1.007 (1.144)	–	–	–	
p(for)	2.55	0.107	3	−2.494 (1.176)	–	–	–	−0.780 (0.637)	
Opossum									
p(.)	0.00	0.545	2	−0.621 (0.324)	–	–	–	–	
p(pina)	3.08	0.117	3	−0.543 (0.402)	−0.207 (0.660)	–	–	–	
p(for)	3.12	0.115	3	−0.612 (0.324)	–	–	–	0.080 (0.339)	
p(tnp)	3.17	0.112	3	−0.628 (0.335)	–	−0.025 (0.291)	–	–	
p(dist)	3.17	0.112	3	−0.621 (0.323)	–	–	0.039 (0.417)	–	
Notes.

Models presented make up the 95% confidence set, where Δi is AICc difference, ωi is the Akaike weight, and K is the number of model parameters. Model covariates were used as a constant detection set for occurrence models for species that did not exhibit the p(.) as the top ranking model (tayra and coati).

Covariates

pina the binomial term to identify large-scale pineapple production within the site buffer

tnp the total number of patches within the buffer

dist the linear distance (km) to the nearest village

for the total percent of forest cover (primary, secondary, and tree plantations) within the site buffer

From the occurrence models, no covariates that we examined explained significant changes in mesopredator occurrence at the study sites (Table 4). The top-ranking models for tayra and raccoon suggested a negative influence of forest cover on both species’ occurrence, but were highly variable. The constant occurrence model was top-ranking for both coati and opossum.

Table 4 Results for occurrence.

Selected top models and untransformed coefficients of habitat variable effects on occurrence models (Ψ) for mesopredators from camera trap surveys in the San Juan - La Selva Biological Corridor, Costa Rica, 2009–2010.

				Untransformed coefficients of covariates (SE)	
Species
model	Δi	ωi	K	Intercept	Forest	Maquenque	Total number of
patches	Pineapples	Distance	
Tayra										
Ψ(for)	0.00	0.432	4	1.568 (2.074)	−2.874 (2.709)	–	–	–	–	
Ψ(Maq)	1.96	0.162	4	1.820 (1.333)	–	−2.875 (1.849)	–	–	–	
Ψ(tnp)	2.14	0.148	4	1.485 (1.365)	–	–	1.507 (1.478)	–	–	
Ψ(pina)	2.83	0.105	4	−0.464 (1.393)	–	–	–	2.313 (1.955)	–	
Ψ(dist)	3.35	0.081	4	1.905 (1.651)	–	–	–	–	0.393 (1.144)	
Coati										
Ψ(.)	0.00	0.487	3	0.532 (0.569)	–	–	–	–	–	
Ψ(for)	2.63	0.131	4	0.679 (0.676)	−0.678 (0.673)	–	–	–	–	
Ψ(Maq)	2.77	0.122	4	0.055 (0.727)	–	1.247 (1.307)	–	–	–	
Ψ(tnp)	2.95	0.111	4	0.528 (0.579)	–	–	0.593 (0.667)	–	–	
Ψ(dist)	3.74	0.075	4	0.544 (0.577)	–	–	–	–	0.162 (0.576)	
Raccoon										
Ψ(for)	0.00	0.623	3	−1.940 (1.218)	−2.052 (1.203)	–	–	–	–	
Ψ(.)	2.52	0.177	2	−1.129 (0.729)	–	–	–	–	–	
Ψ(pina)	4.64	0.061	3	−1.845 (1.115)	–	–	–	1.444 (1.458)	–	
Ψ(tnp)	4.69	0.060	3	−1.270 (0.789)	–	–	0.699 (0.718)	–	–	
Opossum										
Ψ(.)	0.00	0.456	2	0.276 (0.569)	–	–	–	–	–	
Ψ(tnp)	2.11	0.159	3	0.287 (0.594)	–	–	−0.600 (0.604)	–	–	
Ψ(for)	3.07	0.098	3	0.278 (0.573)	−0.198 (0.600)	–	–	–	–	
Ψ(Maq)	3.08	0.098	3	0.118 (0.757)	–	0.349 (1.133)	–	–	–	
Ψ(pina)	3.13	0.096	3	0.381 (0.743)	–	–	–	−0.258 (1.138)	–	
Notes.

Models presented make up the 90% confidence set, where Δi is AICc difference, ωi is the Akaike weight, and K is the number of model parameters. Coefficients are in logit space and relate to standardized covariates.

Covariates

for the total percent of forest cover (primary, secondary, and tree plantations) within the site buffer

Maq the binomial term for sites within the proposed Maquenque National Park

tnp the total number of patches within the buffer

pina the binomial term to identify large-scale pineapple production within the site buffer

dist the linear distance (km) to the nearest village

Discussion

No large cats were photographed during the surveys and only a single coyote was photographed at one site (Cove et al., 2013). Local interviews and cattle depredation were evidence that large cats occur in the corridor; however, the sparse records suggest rarity and precluded the use of these presence/absence data as model covariates. Therefore, we were unable to assess the impact and influence of these top predators on the four mesopredator species through trophic interactions, but the observed detection rates in our surveys are higher than other Neotropical studies with intact top predator communities (Tobler et al., 2008).

Landscape changes did affect detection parameters for the tayra and coati. Because camera traps operate 24-hr per day, heightened detection corresponds with increased local abundance or increased localized activity of mesopredators as influenced by landscape covariates. Pineapple production had a strong positive influence on the detection probability of the tayra. This is most likely an effect of the additional food resources from pineapple production leading to higher local tayra abundance in pineapple-forest edge habitats. The fruits not only provide direct food resources to tayras but other food resources may be indirectly provided from pineapple pests including small rodents, insects, and nesting birds (Presley, 2000). Furthermore this relationship may also be a consequence of concentrated foraging activities within smaller forest patches that commonly occur in the fragmented landscapes associated with pineapple plantations (Cove et al., 2013). Pineapple production also had a positive influence on coati detection, but the effect was less pronounced. Specifically, this suggests that coati abundance is also influenced by agricultural food resources, similar to those associated with tayras, provided from pineapple production. The effect was opposite for raccoons and opossums suggesting that pineapple production has a negative but weak influence on their detection. This result may be an artifact of limited raccoon detections or avoidance of areas of high use by tayras and coatis.

Forest cover had a negative effect on detection probability of coatis. Such an effect suggests that coatis, which occur in large groups, concentrate their foraging activities in small forest patches, making them more easily detectable. This relationship was similar for the detection of tayras and raccoons in the study area. The consequence of such concentrated foraging activities in small forest patches and forest edges has also been shown to be responsible for exposing nesting song birds to increased predation risk (Cove, Niva & Jackson, 2012; Dijack & Thompson, 2000; Donovan et al., 1997). However, none of the habitat covariates examined in this analysis were significant predictors of mesopredator occurrence. Although the coefficients for habitat generally agreed with a priori expectations that increased forest cover would have a negative but variable influence on mesopredator occurrence, lack of significant covariate effects suggests that the broad range of habitats used by these predators could have drastic consequences for nesting song birds, small mammals, and herpetofauna if fragmentation and forest loss continues.

Although low detections of large predators made it difficult to provide direct support for the MRH, the rarity of these species most likely plays a role in the distribution and habitat use by mesopredators. More importantly, the compounding factors of increasing human presence, decreasing forest cover, and increasing pineapple production play an important role in mesopredator release and potentially heightened local abundance. Further sampling of mesopredator communities, as well as large predator-specific surveys and avian point count surveys, will elucidate trophic interactions and the risk of predation to migratory and resident song birds.

Supplemental Information

Supplemental Information 1 Raw data

Click here for additional data file.

We thank all the field assistants and lodges that helped with logistics for this research. Special thanks to Daniel Corrales and Panthera—Costa Rica, Finca Pangola, Selva Verde Lodge, and Laguna Lagarto Lodge for their assistance and continued support.

Additional Information and Declarations

Competing Interests

Author Contributions

Animal Ethics

The authors declare there are no competing interests.

Michael V. Cove conceived and designed the experiments, performed the experiments, analyzed the data, contributed reagents/materials/analysis tools, wrote the paper, prepared figures and/or tables, reviewed drafts of the paper.

R. Manuel Spínola, Victoria L. Jackson and Joel C. Saénz conceived and designed the experiments, contributed reagents/materials/analysis tools, reviewed drafts of the paper.

The following information was supplied relating to ethical approvals (i.e., approving body and any reference numbers):

The camera trapping protocol was approved by the University of Central Missouri Institutional Animal Care and Use Committee (IACUC – Permit No. 10-3202).

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
