# Peer review of "The role of fragmentation and landscape changes in the ecological release of common nest predators in the Neotropics"

_PeerJ, doi:10.7717/peerj.464_

## Round 0.1 · original submission · Minor Revisions

Based on the review comments received, I feel that your article could be reconsidered for publication after some revisions.

Reviewer 1 ·

Basic reporting

It is significant to explore the relationship between mesopredators and the habitat. The article revealed the role of fragmentation and landscape changes in the release of common nest predators. It is helpful for human to recognize how our behavior works on the ecological habitat.
But I am very concerning the research site selection. How did authors make a decision to choice this area. If the observation place changed does the result also change?

Experimental design

I think the experimental framework is rational. However, the study should choice multiple places and then can prove the result is robust.

Validity of the findings

The findings are logical and meaningful. They are also significant to help human to know the effects of land use.

Additional comments

(1) explain why to select the current areas to investigate the common nest predators under the impacts of fragmentation and landscape changes.
(2) How to prove the findings.

·

Basic reporting

Although I found the article interesting, informative, and relatively easy to read, the article did have a few areas of grammatical, sentence structure, and word choice issues that made these sections a little unclear. I have made specific comments in the article pdf.

Experimental design

No Comments

Validity of the findings

No Comments

---

## Round 0.2 · accepted · Accept

Dear authors,

I am an academic editor of PeerJ. I read your revised manuscript, and found that you have carefully revised your paper according to review comments. So, I am pleased to inform you that your article has been accepted for publication in "PeerJ".

Regards

Jianhua Xu